# Network-Based Analysis to Identify Hub Genes Involved in Spatial Root Response to Mechanical Constrains

**DOI:** 10.3390/cells11193121

**Published:** 2022-10-04

**Authors:** Anastazija Dimitrova, Gabriella Sferra, Gabriella Stefania Scippa, Dalila Trupiano

**Affiliations:** Department of Biosciences and Territory, University of Molise, 86090 Pesche, Italy

**Keywords:** poplar, functional enrichment analysis, proteomics, bending, gene ontologies, clusters

## Abstract

Previous studies report that the asymmetric response, observed along the main poplar woody bent root axis, was strongly related to both the type of mechanical forces (compression or tension) and the intensity of force displacement. Despite a large number of targets that have been proposed to trigger this asymmetry, an understanding of the comprehensive and synergistic effect of the antistress spatially related pathways is still lacking. Recent progress in the bioinformatics area has the potential to fill these gaps through the use of in silico studies, able to investigate biological functions and pathway overlaps, and to identify promising targets in plant responses. Presently, for the first time, a comprehensive network-based analysis of proteomic signatures was used to identify functions and pivotal genes involved in the coordinated signalling pathways and molecular activities that asymmetrically modulate the response of different bent poplar root sectors and sides. To accomplish this aim, 66 candidate proteins, differentially represented across the poplar bent root sides and sectors, were grouped according to their abundance profile patterns and mapped, together with their first neighbours, on a high-confidence set of interactions from STRING to compose specific cluster-related subnetworks (I–VI). Successively, all subnetworks were explored by a functional gene set enrichment analysis to identify enriched gene ontology terms. Subnetworks were then analysed to identify the genes that are strongly interconnected with other genes (hub gene) and, thus, those that have a pivotal role in the bent root asymmetric response. The analysis revealed novel information regarding the response coordination, communication, and potential signalling pathways asymmetrically activated along the main root axis, delegated mainly to Ca^2+^ (for new lateral root formation) and ROS (for gravitropic response and lignin accumulation) signatures. Furthermore, some of the data indicate that the concave side of the bent sector, where the mechanical forces are most intense, communicates to the other (neighbour and distant) sectors, inducing spatially related strategies to ensure water uptake and accompanying cell modification. This information could be critical for understanding how plants maintain and improve their structural integrity—whenever and wherever it is necessary—in natural mechanical stress conditions.

## 1. Introduction

Mechanical stress is a naturally occurring abiotic stimulus largely impacting plant growth and development. The ability to perceive mechanical stress is fundamental to all plants, which typically acclimate to such disturbances via developmental responses that modulate the mechanical properties of load-bearing organs and related tissues. At the organ level, in both stem and root, the reduction of mechanical loads is achieved by a decrease in elongation, alteration in the pattern of lateral root and branch initiation, and modification of tissue flexural rigidity (xylem thickening and cell-wall lignification) through reaction wood (RW) formation [1,2,3,4,5,6,7,8,9]. Furthermore, valuable differences have been observed along the main bent root axis, between the convex (cx) and the concave (cv) sides of three different bent sectors: above bending sector (ABS), bending sector (BS) and below bending sector (BBS) [2,9]. In short, as reported in detail in our previous work [9], the compression forces, notable in BS-cv and the adjacent BBS-cv, are able to induce the formation of RW, characterized by an increased cambial cell number and a xylem thickness as well as higher lignin content.

In ABS-cx and the adjacent BS-cx, the tension forces induce the formation of new lateral roots. This asymmetrical structural organization represents the best engineering solution to counteract mechanical deformation, due to which anatomical structure was redesigned to reinforce the taproot, redirect growth toward the vertical direction, and guarantee effective water transport in deforming conditions.

Auxin was found strictly associated with the promotion of cell division and unidirectional formation of RW toward BS-cv whereas, according to the authors of [10], it preceded lateral root formation on BS-cx. Furthermore, cytokinins (CK), in particular zeatin-type, act as central factors opposing gravitropism while abscisic acid (ABA) was assumed to be involved in the water stress response imposed by a bend-induced deforming condition. Several functional proteins, such as annexin, ankyrin, nucleotide diphosphate kinase (NDPK), phosphodiesterase, peroxidase, ara4-interacting protein, ROS signalling, F-actin binding, and Ca^2+^ channel activities, were also found to be asymmetrically modulated in the different bent sides and sectors [3,4,9].

The force perception at the tissue level also triggered the activation of specific signal pathways that, in turn, induced structural variations, in particular, in the vascular cambium and the surrounding developing phloem and xylem tissues. In this process, again, auxin and CK acted directly as spatial regulators of cambial activity, enhancing the cell division rate, and conferring key positional information to the cells for differentiation and RW initiation. ABA was confirmed to represent the main actor regulating the wood hydraulic conductivity improvement and mechanical rigidity in compressed deforming conditions, maintaining RW formation over time [8]. Thus, it is very clear that the different bent sectors, sides and tissues, sense and respond to mechanical signals, along with specific phytohormone signatures, which act both as a morphogen (a compound that confers positional information to cells in a concentration-dependent manner) and a transmitter or mediator of mechanical signals that induce the development of specific cell types, states, and responses.

Despite a large number of targets that have been proposed for the asymmetrical sides’ responses, understanding the comprehensive and synergistic effect of the antistress pathways is still lacking [11,12,13,14]. Recent progress in the bioinformatics area has the potential to fill these gaps through the use of in silico studies to model and evaluate the complexity of biological systems. A variety of systems biology approaches based on the analysis of protein–protein interactions inferred from databases, from genomic or transcriptomic data, or from experimental approaches has been developed to identify functionally related groups of genes involved in coordinated signalling pathways and molecular activities [15,16,17]. These approaches have been widely used to investigate biological functions and pathway overlaps and to identify promising targets in various fields, from anticancer treatments to plant responses [18,19,20,21,22,23], under the general rule of considering the subnetworks composed of candidate proteins together with their neighbours [24]. Thus, in the present work, for the first time, a comprehensive network-based analysis of proteomic signatures was used to identify functions and pivotal genes involved in the coordinated signalling pathways and molecular activities that asymmetrically modulate the response of different bent root sectors and sides. This could lead to the understanding of how plants maintain and improve their structural integrity—whenever and wherever it is necessary—to ensure stability and water transport in natural mechanical stress conditions.

## 2. Materials and Methods

### 2.1. Data Source

The proteomic datasets of *Populus nigra* woody bent root were obtained from our previous study [9]. Briefly, to simulate mechanical perturbations, the *Populus nigra* taproots, freed from all laterals, were bent to a right-angle curved steel net of 90°; as a control, taproots were tied to a straight steel net (Appendix A). Bent and control seedlings were grown for 6 months under a controlled water regime, natural photoperiod, and temperature. Three different sectors of the roots, each 5 cm long, were determined and sampled: (1) the above bending sector located just above the bending zone (ABS); (2) the bending sector of the maximum radical bend (BS); and (3) the below bending sector located just below the bending zone (BBS). In all three sectors, the convex (cx) and concave (cv) sides were separately considered and collected. Successively, as previously reported [2], the total proteins were extracted from taproot samples following the phenol protocol [25], separated by two-dimensional gel electrophoresis (2-DE) and analysed by PDQuest software (Bio-Rad, Hercules, CA). From the analysis, 66 protein spots were differentially represented, considering *p* < 0.01 as the statistical Student’s *t*-test analysis level and an absolute two-fold change in normalized spot densities. These spots were excised from gels, identified by mass spectrometry analysis, and functionally classified according to Bevan [26] (Appendix A).

### 2.2. K-Means Analysis and Subnetwork Identification

The abundance of the 66 differentially represented protein spots was normalized according the formula:*𝑋_normalized_* = log_2_ (*x_i_*/*x̄**_protein_*)
where *x_i_* is *i*-th abundance value and *x̄**_protein_* is the mean abundance of a protein [27]. Then, these values were taken to construct the protein abundance profiles (PAPs), which describe the mean abundance of a protein in the different sectors and sides. PAPs were grouped by using k-means analysis with the ‘kmeans’ function from the package *factoextra* [28] in R environment [29] for clustering proteins. Sector-specific PAP peaking clusters were determined by using a score ≥ 0.3 and the related proteins were used in the following subnetwork identification. Specifically, a high-confidence set of protein–protein interactions (score ≥ 0.7) [30] was selected from *Populus trichocarpa* interactome available in the STRING database [31]. These interactions were visualized and modelled into a network by Cytoscape version 3.8.2, an open software for visualization, integration, and analysis of biomolecular networks [32]. Cluster-related subnetworks were identified by highlighting the proteins of each cluster into the network and selecting them together with their direct interactions and interactors (first direct neighbours).

### 2.3. Gene Set Enrichment Analysis

A functional gene set enrichment analysis (GSEA) was performed on cluster-related subnetworks using g: GOSt (g:Profiler) tool [33] in order to identify enriched gene ontology (GO) terms across the three domains: biological process (GO-BP), molecular function (GO-MF), and cellular component (GO-CC). [34]. Significant GO-BP, GO-MF, and GO-CC terms (Bonferroni adjusted *p* < 0.05) were successively analysed and summarized from a semantic point of view using the REVIGO algorithm by using 0.5 as a cut-off (stringent grouping of similar ontological terms) [35]. Finally, all GO term differences and overlaps among subnetworks were interpretated by Interactivenn [36].

### 2.4. Identification of Hub Genes

Different topological algorithms are available in a Cytoscape plugin, *cytoHubba* (version 0.1) [37], to rank node relevance to the topology of a network. We opted for the use of the “Maximal Clique Centrality” (MCC) algorithm, based on the interconnection between genes and on the identification of the number of maximal cliques to which a node may belong. The MCC algorithm was applied to each cluster-related subnetwork to identify hub genes that are more relevant to the subnetwork topology. The algorithm identifies large clusters of nodes within a network, then scores any node *v* according to the frequency by which it tends to be present in the already identified clusters as defined by the formula:MCC(*v*) = ∑*_C_**_∈_**_S_*_(*v*)_(|*C*|−1)!
where *S*(*v*) is the collection of maximal cliques that contain v and (|*C*|−1)! is the product of all positive integers less than |*C*|. The MCC of node *v* is equal to its degree if there are no connections (edges) between its neighbour nodes. Genes with the highest MCC score tend to codify for essential proteins and for this defined hub gene.

According to the MCC ranking, the top three genes were taken as hub genes and then explored by exploiting database knowledge through UniProt [38], Blast [39], and PopGenie [40].

## 3. Results

A network-based approach was used, for the first time, for a comprehensive functional analysis of total poplar bent root proteomic signatures [9] to identify specific signalling involved in the modulation of the asymmetrical mechanical stress response. In particular, a specific bioinformatic pipeline was designed and implemented to investigate biological functions and pathway overlaps and to identify promising targets of this asymmetry, considering subnetworks composed by candidate proteins together with their neighbours. The steps of the bioinformatic pipeline are summarized in Figure 1 and are fully described in the next paragraphs.

### 3.1. Cluster Analysis of Protein Abundance Profiles

The 66 protein spots, which were differentially represented (Student’s *t*-test *p* < 0.01 and a two-fold change in normalized spot densities) in our previous proteomic analysis [9], were subjected to k-means analysis (Appendix A). Based on their abundance profiles (PAPs), proteins were grouped in six main clusters (I-VI) that characterize different sectors and sides of bent taproot (Table 1 and Figure 2).

In detail, cluster I, showing the highest PAP value in ABS-cx (score: 0.758156; Table 1), grouped twelve proteins (Figure 2A). Cluster II, showing the highest PAP value in BBS-cv (score 0.726657; Table 1), consisted of four proteins (Figure 2B). Cluster III, with the highest abundance in BS-cx (score 0.793721; Table 1) and BBS-cx (score 0.46037; Table 1), grouped eleven proteins (Figure 2C). Cluster IV, with the highest abundance in ABS-cx (score 1.019561; Table 1), BS-cv (score 0.864264; Table 1), and ABS-cv (score 0.464451; Table 1), consisted of three proteins (Figure 2D). Cluster V grouped 16 proteins with the highest abundance profile in ABS-cx (score 0.44414; Table 1) and ABS-cv (score 0.422867; Table 1) (Figure 2E). Finally, Cluster VI, with the highest PAP value corresponding to BBS-cx (score 0.300643; Table 1), consisted of 20 proteins (Figure 2F).

### 3.2. Network and Gene Set Enrichment Analysis

To compose specific cluster-related subnetworks, proteins belonging to each cluster (cluster I–VI) were mapped, together with their first neighbours, on a high-confidence set of interactions (*P. tricocharpa* protein–protein interactions (PPIs)) from STRING. Six cluster-related subnetworks (I–VI) were obtained (Appendix A) and subjected to functional gene set enrichment analysis (GSEA). GSEA identified a variable number of enriched gene ontology (GO) terms (Appendix A) across the three domains—biological process (GO-BP), molecular function (GO-MF), and cellular component (GO-CC)—whose redundancy was reduced and summarized by using REVIGO (Appendix A). In detail, a total of 574 enriched GO terms, composed of 339 GO-BPs, 128 GO-MFs, and 107 GO-CCs, were identified and differently presented across the cluster-related subnetworks I–VI (Appendix A).

Of the 339 GO-BPs (165 without overlaps), 91 were shared among different clusters (Table in Figure 3A; Appendix A) and 74 were identified as subnetwork-specific (Table 2; Appendix A).

Of the 128 GO-MFs (77 without overlaps), 28 were shared among different clusters (Figure 3B; Appendix A) and 49 were identified as subnetwork-specific (Table 2; Appendix A).

Of the 107 GO-CCs (59 without overlaps), 26 were shared among different clusters (Figure 3C; Appendix A) and 33 were identified as subnetwork-specific (Table 2; Appendix A). There were no subnetwork-specific GO-CCs in the case of clusters I, II, and VI (Figure 3C).

Exploring the distribution of specific GO terms among clusters, we found that the cluster-related subnetwork I (ABS-cx-specific) was characterized by 11 GO-BPs related to nucleoside metabolism (*GO:0046940*, *GO:0009123*, *GO:0009144*, *GO:0009199*, *GO:0009142*, *GO:0009201*, *GO:0009058*, *GO:0000461*), the organic substance metabolic process (*GO:1901576*), and the response to oxidative stress (*GO:0006979*) and 3 GO-MFs related to pectate lyase (*GO:0030570*), carbon-oxygen lyase activity acting on polysaccharides (*GO:0016837*), and thioredoxin peroxidase (*GO:0008379*) activities (Table 2; Appendix A).

The cluster-related subnetwork II (BBS-cv-specific) was composed of 2 GO-BPs, associated with the response process (*GO:0050896*, *GO:0042221*) and 13 GO-MFs, mainly related to oxidoreductase/transferase (*GO:0004364*, *GO:0016765*, *GO:0015036*, *GO:0016667*, *GO:0016740*, *GO:0015035*) and binding activities (*GO:1900750*, *GO:0043295*, *GO:0072341*, *GO:1901681*, *GO:0042277*, *GO:0033218*, *GO:0005515*) (Table 2; Appendix A).

The cluster-related subnetwork III (BS-cx- and BBS-cx-specific) was characterized by 23 GO-BPs related to the acetyl-CoA biosynthetic process (*GO:0006086*, *GO:0006085*, *GO:0006084*, *GO:0071616*), nucleoside metabolism (*GO:0034033*, *GO:0033866, GO:0034030*), peptide or hormone or small molecule stimulus (*GO:1901652*, *GO:0071375*, *GO:0032869*, *GO:1901653*, *GO:0032868*, *GO:0043434*, *GO:0044282*), the phosphorus or phosphate metabolic process (*GO:0006793*, *GO:0006796*, *GO:0016310*), the organic compound metabolic process (*GO:0035384*, *GO:1901616*, *GO:0046174*, *GO:0046164*), and the regulation of vacuole organization (*GO:0044088*); 5 GO-MFs with oxidoreductase (*GO:0016624*, *GO:0016860*), NAD binding (*GO:0051287*), phosphopyruvate hydratase (*GO:0004634*), and fructose 1,6-bisphosphate 1-phosphatase (*GO:0042132*) activities; and 13 GO-CCs related to the mitochondrial respirasome machine or structure (*GO:1990204*, *GO:0005746*, *GO:0098803*, *GO:0070469*, *GO:0031975*, *GO:0031967*, *GO:0030964*, *GO:0005747*, *GO:0045271*, *GO:0005743*, *GO:0019866*, *GO:0031966*, *GO:0005740*) (Table 2; Appendix A).

The cluster-related subnetwork IV (BS-cv-, ABS-cv-, and ABS-cx-specific) was represented by 11 GO-BPs, mainly related to molecule transport and localization (*GO:0051179, GO:1902600*, *GO:0006812*, *GO:0034220*, *GO:0098660*, *GO:0098662*, *GO:0098655*, *GO:0055085*, *GO:0006811*, *GO:0006810*, *GO:0051234*); 18 GO-MFs related to ATPase-coupled ion transmembrane transporter activity (*GO:0046961*, *GO:0044769*, *GO:0042625*, *GO:0009678*, *GO:0019829*, *GO:0015078*, *GO:0042626*, *GO:0015399*, *GO:0016887*, *GO:0022853*, *GO:0022890*, *GO:0008324*, *GO:0022804*, *GO:0015318*; *GO:0015075*, *GO:0022857*, *GO:0005215*, *GO:0008553*); and 9 GO-CCs associated with proton-transporting V-type ATPase (*GO:0016469*, *GO:0033176*, *GO:0033177*, *GO:0033179*, *GO:0033180*, *GO:0005773*, *GO:0005774*, *GO:0031090*, *GO:0098588*) (Table 2; Appendix A).

The cluster-related subnetwork V (ABS-cx- and ABS-cv-specific) was characterized by 25 GO-BPs related to proteasomal ubiquitin-independent or -dependent protein degradation (*GO:0010499*, *GO:0043632*, *GO:0030163*, *GO:0043161*, *GO:0010498*, *GO:0044265*, *GO:0006511*, *GO:0044257*, *GO:0051603*, *GO:0019941*, *GO:0006508*, *GO:0044267*, *GO:0019538*, *GO:0009057*, *GO:0044248*, *GO:0043933*), the transcription initiation process (*GO:2000144, GO:0045899, GO:0060260*, *GO:0045898*, *GO:0060261*), and the organonitrogen compound metabolic process (*GO:0006807*, *GO:1901565*, *GO:0044238*, *GO:0071704*); 5 GO-MFs with peptidase activities *(GO:0004298*, *GO:0070003*, *GO:0004175*, *GO:0008233*, *GO:0036402*); and 11 GO-CCs associated with the regulation of the proteasome particle and complex (*GO:0000502*; *GO:1905369*, *GO:1905368*, *GO:0005839*, *GO:0140535*, *GO:0019773*, *GO:0005838*, *GO:0022624*, *GO:0008541*, *GO:0031597*, *GO:0008540*) (Table 2; Appendix A).

The cluster-related subnetwork VI (BBS-cx-specific) was characterized by 2 GO-BPs related to the chemical stimulus response (*GO:0070887*) and aerobic respiration (*GO:0009060*) and 5 GO-MFs associated with the translation regulator (*GO:0008135*, *GO:0090079*, *GO:0045182*) and malate dehydrogenase (*GO:0030060*, *GO:0016615*) activities (Table 2; Appendix A).

### 3.3. Identification of Subnetwork-Related Hub Genes

The Cytoscape plugin *cytoHubba* was applied (Maximal Clique Centrality (MCC) algorithm) to rank node relevance to the topology of each cluster-related subnetwork (I–VI) and identify genes that are strongly interconnected with other genes in each subnetwork (top three hub genes). The top three hub genes, more relevant to each cluster-related subnetwork topology, were selected and explored through the UniProt [38], Blast [39], and PopGenie [40] databases (Table 3). In particular, three 60s ribosomal protein L5 (POPTR_0013s13220, POPTR_0014s17230, and POPTR_0019s13040) were identified as the top three hub genes of cluster I; two glutathione reductase chloroplastic isoform X1 (POPTR_0001s14480 and POPTR_0003s17670) and a probable phospholipid hydroperoxidase glutathione peroxidase (POPTR_0003s12620) for cluster II; three dihydrolipoyl dehydrogenase 2 (POPTR_0008s10700 and POPTR_0010s15200 as chloroplastic isoform and POPTR_0010s16120 mitochondrial isoform) for cluster III; three V-type proton ATPase catalytic subunit A/d2 (POPTR_0008s00560, POPTR_0017s11530, and POPTR_0017s11540) for cluster IV; three proteasome subunit alpha type−6/beta type-2-A (POPTR_0006s14260, POPTR_0008s15530, and POPTR_0016s14640) for cluster V; and a glucose-6-phosphate isomerase 1 chloroplastic (POPTR_0002s10420), an uncharacterized protein LOC7477096 (POPTR_0005s07990), and a phosphoglycerate mutase-like protein 4 (POPTR_0007s11330) for cluster VI. Analysis showed that in all subnetworks, these top three hub genes were connected by interactions and formed triangles.

## 4. Discussion

Previous studies demonstrated that, besides ensuring biomechanical functions, the bent poplar root uses different spatially related strategies to reinforce its structure and maintain water uptake and transport in the three deformed sectors: above bending sector (ABS), below bending sector (BBS), and bending sector (BS). The new cellular geometry imposed by the bend, with cells being slightly stretched on the convex side (cx) and compressed on the concave side (cv), leads to an asymmetric developmental response along the woody root axis [1,4,9]. Indeed, we observed that bent woody root increases xylem thickness, through unidirectional RW formation, on the concave compressed side of BS (BS-cv), and enhances lateral root formation on the convex stretched side of ABS and BS (ABS-cx and BS-cx). According to the authors of [41], the slightly larger cells on the convex-stretched side of the bend act as more efficient auxin sinks that are able to increase lateral root production in ABS-cx [4]. However, in BS-cx, where a low IAA content was found, we hypothesized, as also assumed in *Arabidopsis* bent roots [10], that auxin dynamics preceded and were correlated with curve-dependent lateral root initiation, but with time progression, tension force alone was responsible for lateral root formation at the convex stretched side of the curve. The stress-related anatomical changes in the concave compressed sectors (BS-cv and BBS-cv), expressed through the RW formation, were also due to an auxin-induced increase of cambial activity [8,9]. Besides the role of auxin distribution among different sectors and sides, knowledge of how mechanical forces are spatially assessed and how the signals are later transduced to induce the appropriate responses is still fragmentary. Recent progress in the bioinformatic area has the potential to overcome these shortcomings by using several powerful network-based pipelines that are able to formulate hypotheses and derive biological knowledge [17,22]. Thus, in the present work, the knowledge gained from our proteomic study [9] was extended through an in silico network-based analysis. In particular, a high-confidence set of *P. thricocarpha* protein–protein interactions was selected from STRING and modelled into six subnetworks, selecting the proteins grouped (k-means analysis) according to their abundance profile (PAP) in six clusters (cluster I–VI), together with their first neighbours. The functional analysis of cluster proteins, together with their first neighbours, elucidates commonalities and specificities of the different bent root sectors and sides to adjust their response. It allowed us to derive the sector- and side-related subnetworks together with their functional analysis and concurrently identify related groups of strongly interconnected genes (hub genes) involved in coordinated signalling pathways and molecular activities along the bent root axis.

The results of our network-based analysis reveal new insight on spatially related strategies, commonly or specifically activated in the different bent root sectors (ABS, BS, and BBS) and sides (cx and cv). In particular, ABS-cx was characterized by factors (GO terms grouped in cluster-related subnetwork I) strictly linked to and essential for plant development and responses to mechanical stimuli [42], primarily related to the nucleotides metabolism (GO-BP) and pectate lyase and thioredoxin activities (GO-MF). Furthermore, here, the 60S ribosomal protein L5 (POPTR_0013s13220, POPTR_0014s17230, and POPTR_0019s13040) resulted as the top three hub genes.

Among different signals involved in the mechanical stress response, a nucleoside diphosphate kinase (NDPK), with phosphodiesterase, peroxidase, ROS signalling, F-actin binding, and Ca^2+^ channel activities, already found over-represented in ABS-cx [9], could show a wide range of functions in this sector, such as control of lateral root development [43,44,45,46]. Thioredoxin, another key player in plant cell redox homeostasis, seems able to modulate the functions of target proteins, such as calcium-sensing proteins, particularly on the plasma membrane. In particular, in *A. thaliana* redox regulation of AtCPK21 by thioredoxin was reported in response to external stimuli, playing a pivotal role by amplifying and diversifying the action of Ca^2+^-mediated signals [47]. A rapid increase of the cytosolic Ca^2+^ concentration on the convex side could be linked to the production of apoplastic ROS to rigidify and strengthen the cell wall through oxidative cross-linking of cell wall components [48]. Furthermore, it is important to induce the recruitment of new lateral roots toward the convex root curved region [4,49]. Pectate lyase could also have an important role in the control of new lateral root emergence in ABS-cx as a major component of cell wall remodelling thought pectate/polysaccharide cell wall degradation [50,51]. A general comprehensive protein quality controlling system seems to also be important in ABS-cx. Indeed, the hub genes, all 60S ribosomal protein L5, indicate that an efficient synthesis of new proteins is important to maintain protein homeostasis and to guarantee and adapt the response machinery [52,53]. However, as suggested by subnetwork V, the degradation of misfolded targeted protein seems to be necessary in both ABS-cx and ABS-cv for ensuring the plant plasticity [54,55]. Here, all GO terms (both as BP, MF, and CC) were primarily related to the protein destination together with the top three hub genes (POPTR_0006s14260, POPTR_0008s15530, and POPTR_0016s14640, characterized as ‘proteasome subunit α and β’) (Table 2). Furthermore, it could be increased in the presence of phytohormones, such as abscisic acid (ABA) and cytokinins (CKs) [56], previously found highly accumulated in both sides of ABS [9]. ABA is considered to be the main signalling molecule that activates the adaptive response to osmotic stress [57,58], also under mechanical stress [9]. Cytokinins take on the role of the central antigravitropic determinant in the organ bending [7] alongside an antagonistic cytokinin–auxin relationship [7,59].

Besides this generic response, some other specific GO-BP terms characterized ABS-cx and ABS-cv, together with BS-cv. These GO terms, related to subnetwork IV, were mainly associated with energy-dependent transmembrane transport and were directly correlated with the top three hub genes, POPTR_0008s00560, POPTR_0017s11530, and POPTR_0017s11540, characterized in predictive studies as ‘V-type proton ATPase catalytic’ (V-type ATPase) (Table 2). The V-ATPase is a house-keeping enzyme important for maintaining cytosolic ion homeostasis and cellular metabolism: using the energy released during cleavage cytosolic ATP to pump protons into the vacuolar lumen, thereby creating an electrochemical H^+^-gradient which is the driving force for a variety of transport events of ions and metabolites. Under environmental stress conditions, the V-ATPase functions as a stress response enzyme that moderates expression changes. In particular, the mechanosensitive ion channels present on the plasma membranes are suggested to generate electric action potentials (APs) that propagate on a short distance from cell to cell along with the plasma membrane network and through plasmodesmata (over a longer distance), inducing modifications of cell walls and alterations of microtubule dynamics [60]. Plant APs are widely recognized to incorporate changes in Ca^2+^ as a component of the propagation potential affecting cytosolic pH and ROS generation. These ionic and chemical intermediates, in turn, clearly have profound effects within the cell and on long-term developmental and adaptive processes that rely on modulated gene expression. Thus, the V-ATPase itself might be involved in the process of AP as a component of Ca^2+^-dependent signal transduction chains in ABS-cx, ABS-cv, and BS-cv. Considering the placement of the three sectors on the main bent root axis—the sectors are opposing in the case of ABS-cx and ABS-cv, adjacent in the case of ABS-cv and BS-cv, and not directly in contact in the case of ABS-cx and BS-cv—as well as the significant differences in the anatomical, morphological, and phytohormonal changes among these sectors [9], we can assume that V-type ATPase is involved in a “continuum” response to mechanical stress. In detail, according to the squeeze cell hypothesis [61], mechanical stress could be supposed to modulate V-ATPase, causing ion fluxes to incorporate changes in Ca^2+^, which propagate from cell to cell, reaching cells far away from the maximum of the stimuli. Thus, compression forces sensed on BS-cv is also hypothesized to transmit—from cell to cell—the mechanical stimulus toward ABS-cx, using ABS-cv as a “bridge” to induce new lateral root formation and a new root architectural configuration with improved anchorage features. The connection of these distal portions could require short-distance chemical and electrical signalling [62,63] and hydraulic pulse, along with the plasma membrane network [64] and plasmodesmata over a longer distance. It could also involve the “meristematic connectome”, a recently hypothesized physical cellular network for rapid communication through the plant body’s distant compartments [65].

However, the magnitudes and intensities of mechanical stimuli lead to a customized plant response that varies in compliance with the cell type, developmental stage, and environmental status [66,67]. Indeed, in compressed conditions, such those revealed in BBS-cv, they develop reinforced and lignified secondary wall thickenings to cope with the negative water potential and prevent cellular collapse [68]. Here, ROS signatures induced by compression forces are hypothesized to generate these other stimulus-specific signals.

As the current authors previously found in [9], in BBS-cv, ROS act downstream of auxin in the gravitropic response and tissue reinforcement through a higher degree of lignification. Present findings also identified BBS-cv to be associated with the ROS signature in terms of GO-BP terms (grouped in the cluster-related subnetwork II) associated with oxidoreductase/transferase activity, and the top three hub genes, identified as glutathione reductase or glutathione peroxidase (POPTR_0001s14480, POPTR_0003s12620, and POPTR_0003s17670). The ROS signature is surveyed by the profound ROS gene network, which involves many components, to maintain their homeostasis [69,70,71]. As a member of glutathione family, it is the major antioxidant with a noted response to a range of abiotic stresses (salinity, drought, extreme temperatures, and metal toxicity), able to reduce the oxidative stress, protect the plasma membrane, and prevent lipid peroxidation [72]. As part of the oxido-reduction proteins, POPTR_0003s12620 exhibited a continuous abundance increase during early secondary growth of the poplar stem [73]. In *Arabidopsis,* glutathione activity has been shown to induce a higher level of ABA as part of the response to drought [72] and has been involved in auxin crosstalk, both part of the root architecture modulation under stress conditions [74,75], which is additionally reinforced by our previous findings [9].

While on one hand, compressed compartments may be strictly connected to ROS signatures in resemblance to the Ca^2+^ signal, on the other hand, high tension forces perceived in BBS-cx [9] seem to mainly impact energy and metabolic processes. BBS-cx was characterized (cluster-related subnetwork VI) by GO-BPs associated with ‘cellular response to chemical stimulus’ and ‘aerobic respiration’ together with the GO-MF domain related to ‘L-malate dehydrogenase activity’ and ‘malate dehydrogenase activity’ and the related top three hub genes (POPTR_0002s10420, ‘glucose-6-phosphate isomerase 1′, POPTR_0005s07990, ‘uncharacterized protein LOC7477096′, and POPTR_0007s11330 ‘phosphoglycerate mutase-like protein 4′). Malate dehydrogenase is involved in central metabolism and redox homeostasis between organelle compartments [76], playing major roles in reductant export and thus regulating redox and hormone levels under stress conditions [77]. Glucose-6-phosphate isomerase serves in the oxidative pentose phosphate pathway (OPPP), which is involved in the early response to various abiotic stresses [78,79] and the same role can be attributed to the gene POPTR_0002s10420 in the case of mechanical stress. This gene was also identified as part of the secretory carrier-associated membrane proteins in wood formation in poplar [80] and one of the proteins present in the energy metabolism pathway during the dormancy release stage in poplar [81]. POPTR_0007s11330 was identified as ‘probable phosphoglycerate mutase’ in poplar roots as one of the genes associated with non-structural carbohydrate storage. POPTR_0005s07990 has no mention in the literature; however, it has been associated with 2 GO-terms, *GO:0006095* (glycolysis) and *GO:0016868* (intramolecular transferase activity, phosphotransferases) via PopGenie, which elucidates similar types of activity.

Additionally, BBS-cx showed some similarities with the adjacent sector, BS-cx (also subjected to high tension forces [9]), sharing GO-BP terms assigned to cluster-related subnetwork III and classified most proteins under energy and metabolism. The top three hub genes of this cluster—POPTR_0008s10700, POPTR_0010s15200, and POPTR_0010s16120—have been identified as ‘dihydrolipoyl dehydrogenase isoforms’. Although for all three genes, the literature is scarce, POPTR_0010s15200 is one of the genes associated with poplar early stem development, whose abundance decreases during stem lignification due to remodulation of carbohydrate and energy metabolism [73]. As part of the lipid metabolism pathway, POPTR_0010s16120 (named lipoamide dehydrogenase) was detected and showed an increase during the progression of poplar through dormancy-release stages, when a surplus of energy is also necessary [81]. Considering that both BS-cx and BBS-cx are characterized by higher tension forces, we can hypothesize that they require more energy and the activation of metabolic pathways to optimize the water transport via osmotic adjustment, such as that in BS-cx, is also amplified through the new lateral root formation [9]. Furthermore, they can take part in the main mechanisms for reductive potential energy and regulation of osmotic potential as well as turgor on the opposite compressed sides (BS-cv and BBS-cv). Indeed, in BS-cv and BBS-cv, it has been hypothesized that solutes might move radially along the ray cell walls, enter the embolized xylem conduits, and increase the solute concentration of the residual water within them, thus promoting xylem refilling by altering osmoticum, as has been shown to occur during drought [82,83].

## 5. Conclusions

The network-based pipelines that were applied to proteomic signatures of poplar woody bent taproot confirmed that the convex and concave sides of three bent root sectors (ABS, BS, and BBS) use different strategies to counteract mechanical stress. Specific functions and pivotal genes involved in these coordinated signalling pathways and molecular activities, which asymmetrically modulate the spatially related response, were identified and summarized in the model presented in Figure 4.

In particular, mechanical stimulus was confirmed to induce specific spatially related signalling, along which Ca^2+^ could act on ABS-cx as a signal to translate the mechanical forces inherent in growth to a developmental response in the roots through (i) rapid ROS production to the apoplast, (ii) the regulation of the stress-related protein machine (through trade-off between proteins synthesis and degradation) and, finally, (iii) new lateral root formation (to ensure plant anchorage and water availability). Instead, in BBS-cv, the ROS signature and related modulation of glutathione antioxidant forms serve as part of a gravity-induced bending response that triggers lignin formation.

New insight regarding the response coordination between distant bent root portions (sectors) to induce spatially related strategies was also obtained. In detail, compression forces sensed on BS-cv is supposed to transmit—from cell to cell—the mechanical stimulus toward ABS-cx, using ABS-cv as a “bridge” to induce new lateral root formation and a new root architectural configuration with improved anchorage features. This is supported by our observations regarding the ‘dispersed’ V-type ATPase, which, after being activated by high compressed forces in BS-cv, could cause ion fluxes to incorporate changes in Ca^2+^ that propagate from cell to cell and to reach far away from the maximum of stimuli. BS-cx and BBS-cx also seem to be highly connected to the corresponding opposite side (BS-cv and BBS-cv) to guarantee solute-induced xylem water refilling. The communication between these portions is supposed to engage short distance signals, such as chemical and electrical signalling and plasma membrane hydraulic pulse or plasmodesmata and meristematic connectome to cover long distances and adjust the root body to its surrounding environment. This elucidates a complex interplay between signals and responses that involve downstream effects, effectors, changes in cell adhesion and communication properties, which poses almost unknown and ongoing challenges for the research community to date. Further research focused on the early events in the root bending response, with closer sampling time points, would allow for insight into the mechanical signal transduction pathways involved in the extra- and intracellular space to elucidate how plant tissues are organized through cell–cell communication along with the influences from and to neighbouring and distant cells.

## Figures and Tables

**Figure 1 cells-11-03121-f001:**
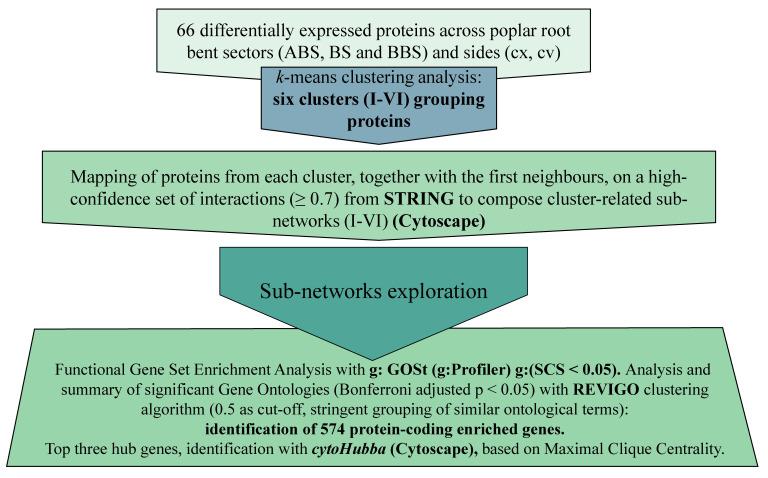
Summary of bioinformatic pipeline. The 66 protein spots found differentially represented among bent sectors and sides [9] were used to perform the network-based analysis. In particular, all protein spots were grouped according to their abundance profile patterns (k-means analysis) and mapped, together with their first neighbours, on a high-confidence set of interactions from STRING to compose specific cluster-related subnetworks (I–VI). Successively, all subnetworks were explored by a functional gene set enrichment analysis to identify enriched gene ontology (GO) terms across the three domains—biological process (GO-BP), molecular function (GO-MF), and cellular component (GO-CC)—and summarized by REVIGO. Subnetworks were then analysed by *cytoHubba* (Maximal Clique Centrality algorithm) to identify the top three hub genes strongly interconnected with other genes.

**Figure 2 cells-11-03121-f002:**
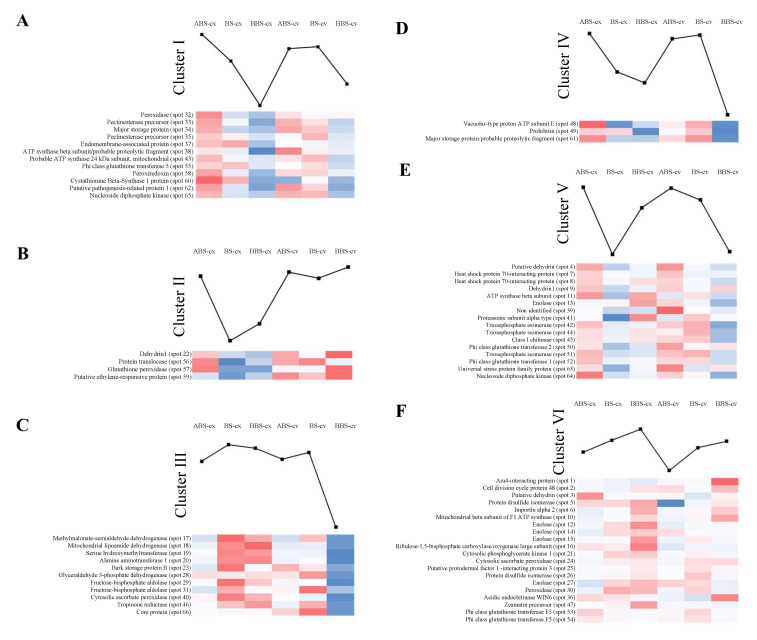
Cluster analysis on protein abundance profiles (PAPs). Subfigures (**A**–**F**), corresponding with the six clusters (I–VI) show the distribution heatmaps of the 66 differentially represented protein spots according to their PAP (k-means analysis). Line charts show the average of PAPs calculated as the mean value of all cluster-related proteins. ABS, above bending sector; BS, bending sector; BBS, below bending sector; cx, convex side; cv, concave side.

**Figure 3 cells-11-03121-f003:**
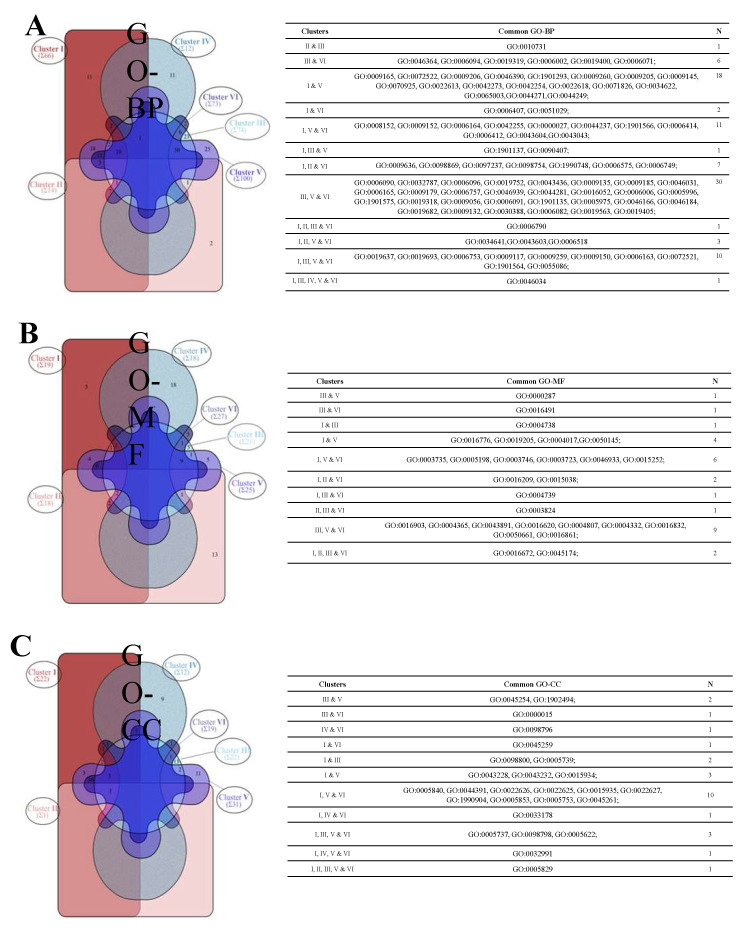
Distribution of GO terms in the cluster-related subnetwork (I–VI). GO term differences and overlaps among subnetworks were interpretated by Interactivenn [36]. Venn diagram (Edward’s style) shows the number of common or specific GO-BPs among subnetworks (I–VI) across the three domains, GO-BP (panel **A**), GO-MF (panel **B**), and GO-CC (panel **C**). Common GO-BP, GO-MF, and GO-CC terms of panel (**A**–**C**) are listed in the Table, respectively. The list of all specific GO terms is reported in Table 2.

**Figure 4 cells-11-03121-f004:**
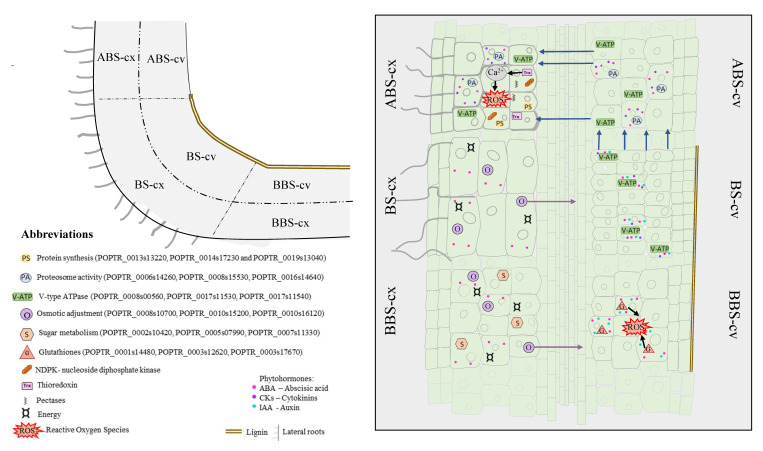
Schematic illustration of network and hub genes involved in spatial root response to mechanical constraints. Representative images summarize processes and pathways that characterize the different bent root sectors and sides. More details provided in the Conclusion section. ABS, above bending sector; BS, bending sector; BBS, below bending sector; cx, convex side; cv, concave side.

**Table 1 cells-11-03121-t001:** k-means analysis. The 66 protein spots, which are differentially represented in poplar bent root, are grouped by k-means analysis in six main clusters (I–VI), characterizing one or more bent root sectors and sides. Sector-specific clusters are determined by using a score ≥ 0.3 (in bold) and the related proteins were used in the following subnetwork identification. ABS, above bending sector; BS, bending sector; BBS, below bending sector; cx, convex side; cv, concave side.

	Bent Root Sectors/Sides						Cluster Size (N° Proteins)	Withinss (Within Cluster Sum of Square)
	ABS-cx	BS-cx	BBS-cx	ABS-cv	BS-cv	BBS-cv		
**Cluster I**	**0.7581557**	−0.29911	−2.06396	0.191851	0.266578	−1.21004	12	36.9506
**Cluster II**	−0.1136721	−6.2065	−4.59749	0.250547	−0.32258	**0.729957**	4	42.15465
**Cluster III**	−0.7552579	**0.793721**	**0.46037**	−0.56888	0.069973	−6.89288	11	38.7308
**Cluster IV**	**1.0195613**	−2.84078	−3.94615	**0.464451**	**0.864264**	−7.14603	3	42.71795
**Cluster V**	**0.4441395**	−1.41837	−0.12500	**0.422867**	0.09185	−1.34692	16	65.03039
**Cluster VI**	−0.3790747	−0.02729	**0.300643**	−0.92325	−0.24844	−0.05900	20	137.90456

**Table 2 cells-11-03121-t002:** Cluster-related subnetwork-specific GO terms. List of cluster-related subnetwork (I-VI) -specific gene ontology (GO) terms across the three domains: biological process (GO-BP), molecular function (GO-MF), and cellular component (GO-CC).

Cluster	Specific GO-BP	Specific GO-MF	Specific GO-CC
I	GO:0046940, GO:0009141, GO:0009123, GO:0009144, GO:0009199, GO:0009142, GO:0009201, GO:0009058, GO:0006979, GO:1901576, GO:0000461.	GO:0030570, GO:0016837, GO:0008379.	
II	GO:0050896, GO:0042221.	GO:0004364, GO:0016765, GO:1900750, GO:0043295, GO:0072341, GO:1901681, GO:0042277, GO:0033218, GO:0015036, GO:0016667, GO:0016740, GO:0005515, GO:0015035.	
III	GO:0006086, GO:0006085, GO:0035384, GO:0034033, GO:0006084, GO:0033866, GO:0071616, GO:0034030, GO:0044088, GO:0032889, GO:1901652, GO:0071375, GO:0032869, GO:1901653, GO:0032868, GO:0043434, GO:0044282, GO:0006793, GO:0006796, GO:0016310, GO:1901616, GO:0046174, GO:0046164.	GO:0016624, GO:0051287, GO:0004634, GO:0042132, GO:0016860.	GO:1990204, GO:0070469, GO:0031975, GO:0031967, GO:0005746, GO:0005747, GO:0019866, GO:0098803, GO:0005743, GO:0005740, GO:0031966, GO:0045271, GO:0030964.
IV	GO:0051179, GO:0098655, GO:1902600, GO:0006812, GO:0098660, GO:0034220, GO:0098662, GO:0006811, GO:0055085, GO:0006810, GO:0051234.	GO:0046961, GO:0044769, GO:0042625, GO:0009678, GO:0019829, GO:0015078, GO:0042626, GO:0015399, GO:0016887, GO:0022853, GO:0022890, GO:0008324, GO:0022804, GO:0015318, GO:0015075, GO:0022857, GO:0005215, GO:0008553.	GO:0016469, GO:0033176, GO:0033179, GO:0033177, GO:0033180, GO:0005773, GO:0005774, GO:0098588, GO:0031090.
V	GO:0010499, GO:0043632, GO:0030163, GO:0043161, GO:0010498, GO:0044265, GO:0006511, GO:0044257, GO:0051603, GO:0019941, GO:2000144, GO:0045899, GO:0060260, GO:0045898, GO:0060261, GO:0043933, GO:0006807, GO:0044238, GO:0071704, GO:0006508, GO:0044267, GO:0019538, GO:1901565, GO:0009057, GO:0044248.	GO:0004298, GO:0070003, GO:0004175, GO:0008233, GO:0036402.	GO:0000502, GO:1905368, GO:0031597, GO:0005838, GO:0019773, GO:0022624, GO:0008541, GO:0005839, GO:0008540, GO:1905369, GO:0140535.
VI	GO:0070887, GO:0009060.	GO:0008135, GO:0090079, GO:0045182, GO:0030060, GO:0016615.	

**Table 3 cells-11-03121-t003:** Cluster-specific candidate hub genes. List of the top three hub genes (*cytoHubba*, MCC algorithm) of each cluster-related subnetwork (I–VI), explored through UniProt [38], Blast [39], and PopGenie [40] databases.

Cluster	MCC	*cytoHubba*	UniProt		Blast	PopGenie
			Protein	Gene		
**I**	9.22 × 10^13^	POPTR_0013s13220	N/A	N/A	60S ribosomal protein L5	Potri.013G128600
	9.22 × 10^13^	POPTR_0014s17230	Ribosomal_L18_c domain-containing protein	POPTR_014G174000	60S ribosomal protein L5	Potri.014G174000
	9.22 × 10^13^	POPTR_0019s13040	Ribosomal_L18_c domain-containing protein	POPTR_019G099000	60S ribosomal protein L5	Potri.019G099000
**II**	222,240	POPTR_0001s14480	N/A	N/A	Glutathione reductase, chloroplastic isoform X1	Potri.001G050000
	226,235	POPTR_0003s12620	Glutathione peroxidase	N/A	Probable phospholipid hydroperoxide glutathione peroxidase	Potri.003G126100
	212,160	POPTR_0003s17670	Glutathione reductase	POPTR_003G178200	Glutathione reductase, chloroplastic isoform X1	Potri.003G178200
**III**	9.22 × 10^13^	POPTR_0008s10700	N/A	N/A	Dihydrolipoyl dehydrogenase 2, chloroplastic isoform X2	Potri.008G107600
	9.22 × 10^13^	POPTR_0010s15200	Uncharacterized protein	POPTR_010G142100	Dihydrolipoyl dehydrogenase 2, chloroplastic	Potri.010G142100
	9.22 × 10^13^	POPTR_0010s16120	N/A	N/A	Dihydrolipoyl dehydrogenase 2, mitochondrial OR lipoamide dehydrogenase	Potri.010G151400
**IV**	9.22 × 10^13^	POPTR_0008s00560	V-ATPase 69 kDa subunit	POPTR_008G005000	V-type proton ATPase catalytic subunit A	Potri.008G005000
	9.22 × 10^13^	POPTR_0017s11530	V-type proton ATPase subunit	POPTR_017G079200	V-type proton ATPase subunit d2	Potri.017G079200
	9.22 × 10^13^	POPTR_0017s11540	N/A	N/A	V-type proton ATPase subunit d2	Potri.017G079200
**V**	9.22 × 10^13^	POPTR_0006s14260	Proteasome subunit alpha type	POPTR_006G140400	Proteasome subunit alpha type-6	Potri.006G140400
	9.22 × 10^13^	POPTR_0008s15530	Proteasome subunit beta	POPTR_008G155500	Proteasome subunit beta type-2-A	Potri.008G155500
	9.22 × 10^13^	POPTR_0016s14640	Proteasome subunit alpha type	POPTR_016G139600	Proteasome subunit alpha type-6	Potri.016G139600
**VI**	9.22 × 10^13^	POPTR_0002s10420	Glucose-6-phosphate isomerase	POPTR_002G104000	Glucose-6-phosphate isomerase 1, chloroplastic	Potri.002G104000
	9.22 × 10^13^	POPTR_0005s07990	N/A	N/A	Uncharacterized protein LOC7477096	Potri.005G078100
	9.22 × 10^13^	POPTR_0007s11330	Uncharacterized protein	POPTR_007G040700	Phosphoglycerate mutase-like protein 4	Potri.007G040700

## Data Availability

Not applicable.

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
