# Peer review of "Network-Based Analysis to Identify Hub Genes Involved in Spatial Root Response to Mechanical Constrains"

_cells, 2022, doi:10.3390/cells11193121_

Round 1

Reviewer 1 Report (Previous Reviewer 2)

The manuscript in the current version is much improved and previous comments have been addressed. I accept it in its current form.

Author Response

Reviewer 1 Comments and Suggestions for Authors

The manuscript in the current version is much improved and previous comments have been addressed. I accept it in its current form.

Answer: We thank the Reviewer.

Reviewer 2 Report (New Reviewer)

In this manuscript, a comprehensive network-based analysis of proteomic signatures was used to identify functions and pivotal genes involved in the coordinated signaling pathways/molecular activities which asymmetrically modulate the response of different bent poplar root sectors and sides. The analysis revealed novel information regarding the response coordination, communication, and potential signaling pathways asymmetrically activated along the main root axis. In addition, some of the data indicate that the concave side of bent sector, where the mechanical forces are most intense, communicates to the other (neighbour and distant) sectors inducing spatially-related strategies to ensure water uptake and accompanying cell modification. This information could be critical for understanding how plants maintain/improve their structural integrity- whenever and wherever it is necessary - in natural mechanical stress conditions. I just request the correction of the following points, 

1) Table S4 contains the key data for this paper. Thus, it should be listed as one of the main tables. 

2) Font sizes in Fig. 1, 3, and 6 are too small. In addition, there are no Fig. 4 and 5 in this manuscript. Please revise them properly. 

Author Response

Reviewer 2 Comments and Suggestions for Authors

In this manuscript, a comprehensive network-based analysis of proteomic signatures was used to identify functions and pivotal genes involved in the coordinated signaling pathways/molecular activities which asymmetrically modulate the response of different bent poplar root sectors and sides. The analysis revealed novel information regarding the response coordination, communication, and potential signaling pathways asymmetrically activated along the main root axis. In addition, some of the data indicate that the concave side of bent sector, where the mechanical forces are most intense, communicates to the other (neighbour and distant) sectors inducing spatially-related strategies to ensure water uptake and accompanying cell modification. This information could be critical for understanding how plants maintain/improve their structural integrity- whenever and wherever it is necessary - in natural mechanical stress conditions. I just request the correction of the following points, 

  • Table S4 contains the key data for this paper. Thus, it should be listed as one of the main tables. 

 Answer: As suggested, the Table containing this data has been transferred from Supp.Mat. into the manuscript as ‘Table 3’.

  • Font sizes in Fig. 1, 3, and 6 are too small. In addition, there are no Fig. 4 and 5 in this manuscript. Please revise them properly. 

 Answer: The figure numbering has been revised and corrected. As suggested, the font size has also been increased in the figures.

Reviewer 3 Report (New Reviewer)

The authors propose network-based analysis to identify hub genes involved in spatial root response to mechanical constrains. The topic is interesting, and the research is with adequate soundness and novelty. However, there are some minor problems that should be pointed out.

1. In this paper, the Maximal Clique Centrality (MCC) algorithm was applied to each cluster-related subnetwork to identify hub genes which are more relevant to the subnetwork topology. The authors should explain more details about MCC algorithm to illustrate the proposed method more clearly.

2. The title and contents of Table 2 can be put on the same page.

3. The "Figure 6. Schematic illustration of network/hub genes involved in spatial root response to mechanical constrains" is the 4th figure of this paper, so it should be "Figure 4".

Author Response

Reviewer 3 Comments and Suggestions for Authors

The authors propose network-based analysis to identify hub genes involved in spatial root response to mechanical constrains. The topic is interesting, and the research is with adequate soundness and novelty. However, there are some minor problems that should be pointed out.

  1. In this paper, the Maximal Clique Centrality (MCC) algorithm was applied to each cluster-related subnetwork to identify hub genes which are more relevant to the subnetwork topology. The authors should explain more details about MCC algorithm to illustrate the proposed method more clearly.

 Answer: As suggested, we have added more information about MCC algorithm in the method section. Line 145-156 “The algorithm identifies large clusters of nodes within a network, then scores any node v according to the frequency by which it tends to be present in the already identified clusters as defined by the formula: MCC(v)=∑CS(v)(|C|−1)!

where S(v) is the collection of maximal cliques which contain v, and (|C|-1)! is the product of all positive integers less than |C|. The MCC of node v is equal to its degree if there are no connections (edges) between its neighbor nodes. Genes with the highest MCC score tend to codify for essential proteins and for this defined hub genes.”

 2) The title and contents of Table 2 can be put on the same page.

 Answer: The title and the table have been put on the same page.

 3) The "Figure 6. Schematic illustration of network/hub genes involved in spatial root response to mechanical constrains" is the 4th figure of this paper, so it should be "Figure 4".

 Answer: The figure numbering has been revised and corrected.

This manuscript is a resubmission of an earlier submission. The following is a list of the peer review reports and author responses from that submission.

Round 1

Reviewer 1 Report

In the manuscript entitled: Network-based analysis to identify hub genes involved in spatial root response to mechanical constraints, the authors identify proteins functions and key genes involved in the coordinated signaling pathways/molecular activities which asymmetrically modulate the response (direction/magnitude) of different bent root sectors and sides.

The analysis suggests a new insight regarding the coordination between distant to induce spatially related strategies able to ensure water uptake.

Minor comments:

The abstract section needs to be rewritten; it does not fully show the importance of this work.

 In my opinion manuscript after this minor revision may be accepted for publication in cells.

Reviewer 2 Report

Network-based analysis to identify hub genes involved in spatial root response to mechanical constrains

The manuscript provides network based insight into responsed mechanical-stress that induced protein-protein interaction and changes in root morphology. The current version is of very basic nature and the whole study is based on the initial results of STRING database. It would be interesting to include the molecular interaction features derived from in-silico tools, and comment on the involved domains making the interface region of interaction. The molecular analyses will not only upgrade this manuscript but similarly will provide information for mutagenesis studies to be tested in wet lab setups. The current version only highlights the interaction probability that can be considered an extreme basic version. However, I will suggest the following major/minor improvements.

COMMENTS

         In lines 68-71, authors should name the system biology approaches based on protein-protein interaction.

         In line 85, you mentioned 2- year old plant. However, your source paper shows the age as 22- years old. Correct it.

         You should mention the way the bending etc. was accomplished.

         In lines 88-90, you mentioned about sampled parts. I will suggest providing pictures (supplementary) of the parts as you explained them differentially.

         What are hub genes?

         Methodology section needs proper elaboration, the current version is very abstractive.

         Sentence construction needs to be improved.

Reviewer 3 Report

Demitrova et al used a published proteomic datasets from Populus nigra woody root to identify hub genes implicated in the spatial root response to mechanical constraint.

They authors performed a K-means analysis on a small size dataset: 66 DE proteins in 3 sections and 2 sides. From the method and from Fig1 it is unclear how the authors selected the first neighbors to each of the 66 DE proteins and it is not clear what network centrality metric was used to identify the hub genes. Overall the manuscript is a list of GO terms that are difficult to interpret since the methodology associated to the identification of the clusters and the related hub genes looks weak.

I suggest to better describe the methods and the downstream results. Several graph centrality metrics can be applied to identify interesting nodes and their downstream and upstream interaction. The IGraph R package is very useful in this case.